# A Set of Cell Lines Derived from a Genetic Murine Glioblastoma Model Recapitulates Molecular and Morphological Characteristics of Human Tumors

**DOI:** 10.3390/cancers13020230

**Published:** 2021-01-10

**Authors:** Barbara Costa, Michael N. C. Fletcher, Pavle Boskovic, Ekaterina L. Ivanova, Tanja Eisemann, Sabrina Lohr, Lukas Bunse, Martin Löwer, Stefanie Burchard, Andrey Korshunov, Nadia Coltella, Melania Cusimano, Luigi Naldini, Hai-Kun Liu, Michael Platten, Bernhard Radlwimmer, Peter Angel, Heike Peterziel

**Affiliations:** 1Division of Signal Transduction and Growth Control, DKFZ-ZMBH Alliance, German Cancer Research Center (DKFZ), 69120 Heidelberg, Germany; ekaterina.ivanova@dkfz.de (E.L.I.); t.eisemann@dkfz.de (T.E.); s.lohr@dkfz.de (S.L.); h.peterziel@kitz-heidelberg.de (H.P.); 2Division of Molecular Genetics, German Cancer Research Center (DKFZ), 69120 Heidelberg, Germany; m.fletcher@dkfz-heidelberg.de (M.N.C.F.); p.boskovic@dkfz-heidelberg.de (P.B.); B.Radlwimmer@dkfz-heidelberg.de (B.R.); 3DKTK CCU Neuroimmunology and Brain Tumor Immunology, German Cancer Research Center (DKFZ), 69120 Heidelberg, Germany; l.bunse@Dkfz-Heidelberg.de (L.B.); m.platten@dkfz.de (M.P.); 4Department of Neurology, MCTN, Medical Faculty Mannheim, Heidelberg University, 69120 Heidelberg, Germany; 5Department of Neurology, Medical Faculty Heidelberg, Heidelberg University, 69120 Heidelberg, Germany; 6TRON—Translational Oncology at the University Medical Center of the Johannes Gutenberg University, 55131 Mainz, Germany; Martin.Loewer@TrOn-Mainz.DE (M.L.); Stefanie.Burchard@TrOn-Mainz.DE (S.B.); 7Department of Neuropathology, Heidelberg University Hospital, 69120 Heidelberg, Germany; Andrey.Korshunov@med.uni-heidelberg; 8CCU Neuropathology, German Cancer Research Center (DKFZ), 69120 Heidelberg, Germany; 9Targeted Cancer Gene Therapy Unit, San Raffaele Telethon Institute for Gene Therapy (SR-Tiget), IRCCS San Raffaele Scientific Institute, 20132 Milan, Italy; coltella.nadia@hsr.it (N.C.); cusimano.melania@hsr.it (M.C.); naldini.luigi@hsr.it (L.N.); 10School of Medicine, Vita-Salute San Raffaele University, 20132 Milan, Italy; 11Division of Molecular Neurogenetics, DKFZ-ZMBH Alliance, German Cancer Research Center (DKFZ), 69120 Heidelberg, Germany; L.Haikun@Dkfz-Heidelberg.de; 12Helmholtz Institute for Translational Oncology (Hi-TRON), 55131 Mainz, Germany; 13Department of Neurology, University Hospital and Medical Faculty Mannheim, 68167 Mannheim, Germany; 14German Cancer Consortium (DKTK), DKFZ, 69120 Heidelberg, Germany

**Keywords:** glioblastoma, mouse model, syngeneic cell line

## Abstract

**Simple Summary:**

Glioblastoma (GBM) is a highly aggressive and almost inevitably lethal brain tumor. Animal models for GBM are crucial to study how the tumor evolves in vivo and to test novel treatment options. Most currently available models are based on the transplantation of human GBM cells into mice with a defective immune system. However, this approach does not allow to study the contribution of immune cells to GBM growth and to test immunotherapies. Transplantation of murine GBM cells overcomes this limitation, however, up to now, only a limited number, which mostly do not mimic important characteristics of human GBM, have been available. Via in vivo passaging, we established a set of murine GBM cell lines that (i) can be easily cultivated and further genetically manipulated, (ii) upon transplantation develop tumors with phenotypic and pathological features of human GBM, and (iii) are available to be shared with the scientific community.

**Abstract:**

Glioblastomas (GBM) are the most aggressive tumors affecting the central nervous system in adults, causing death within, on average, 15 months after diagnosis. Immunocompetent in-vivo models that closely mirror human GBM are urgently needed for deciphering glioma biology and for the development of effective treatment options. The murine GBM cell lines currently available for engraftment in immunocompetent mice are not only exiguous but also inadequate in representing prominent characteristics of human GBM such as infiltrative behavior, necrotic areas, and pronounced tumor heterogeneity. Therefore, we generated a set of glioblastoma cell lines by repeated in vivo passaging of cells isolated from a neural stem cell-specific *Pten/p53* double-knockout genetic mouse brain tumor model. Transcriptome and genome analyses of the cell lines revealed molecular heterogeneity comparable to that observed in human glioblastoma. Upon orthotopic transplantation into syngeneic hosts, they formed high-grade gliomas that faithfully recapitulated the histopathological features, invasiveness and immune cell infiltration characteristic of human glioblastoma. These features make our cell lines unique and useful tools to study multiple aspects of glioblastoma pathomechanism and to test novel treatments in an intact immune microenvironment.

## 1. Introduction

Glioblastomas are the most prevalent malignant brain tumors in adults. These World Health Organization (WHO) grade IV tumors are defined by histopathologic features comprising mitotic figures, microvascular proliferation, necrosis, and extensive infiltration of the brain parenchyma, with the latter impairing complete surgical removal and almost inevitably leading to tumor relapse. The current treatment regimen of glioblastoma patients consists of maximal surgical resections combined with chemo- and radiotherapy; however, the median survival of about 15 months post diagnosis has not improved substantially in the recent decades, making glioblastoma one of the deadliest tumors and underscoring the importance of developing novel, more effective treatments [1]. At the molecular level, 90% of glioblastomas harbor wildtype isocitrate dehydrogenase 1 and 2 genes (*IDH1* and *IDH2*) distinguishing them from the remaining 10%, the so-called secondary glioblastomas, with a better prognosis [2]. Our knowledge of glioblastoma molecular pathomechanisms has greatly advanced in the past decade, and it has become widely accepted that *IDH* wildtype (*IDH*^wt^) glioblastomas can be sub-classified into three subtypes: classical, proneural, and mesenchymal [3]. Importantly, the corresponding subtype-specific RNA-expression-based signatures are also influenced by the composition of the tumor microenvironment, including stromal and immune cell populations [4]. Despite these advances, we are still lacking in-vivo preclinical models that reflect the molecular and phenotypic heterogeneity of glioblastoma and recapitulate their complex interaction with the immune cells of the tumor microenvironment. In order to thoroughly understand the biology underlying glioblastoma development and to allow more predictive preclinical testing of new therapies, those glioblastoma animal models are urgently needed.

Syngeneic models use the transplantation of murine glioma cells into animals with identical genetic background and constitute a promising approach to investigate GBM biology in an immunocompetent context. Currently, there are few established murine glioblastoma cell lines available, including GL261, GL26, CT-2A, SMA-560, and 4C8 [5]. The GL261 cell culture was established in the mid 1990s from the GL261 brain tumor mouse model, which was originally induced in 1939 by intracranial injection of 3-methylcholanthrene and maintained by serial transplantation in syngeneic mice. These cells have been thoroughly characterized and are by far the most extensively used to date [6]. When injected orthotopically, GL261 tumors exhibit a bulky growth pattern [7], forming a nodular mass without considerable infiltration of tumor cells into the brain parenchyma. Additionally, these tumors rarely, if ever, exhibit necrosis, which in addition to the invasive growth, is a notable characteristic of human glioblastoma. Moreover, GL261 cells are molecularly characterized by an activating Kras mutation, which is almost never observed in human GBM [3].

Based on the observation that the two tumor suppressor genes *p53* and *Pten* are frequently inactivated in human GBM [3], we have previously generated a GBM model characterized by the neural stem cell-specific deletion of these two genes [8]. This genetic model recapitulates human GBM features such as aggressive infiltration into the brain parenchyma, intratumoral hemorrhages, and necrosis.

With the aim to establish more authentic syngeneic GBM models, we generated a panel of glioblastoma cell lines from tumor cells of this genetic model [8]. Upon orthotopic transplantation into syngeneic hosts, these cell lines formed high-grade GBM that faithfully recapitulated key characteristics of human glioblastoma. Additionally, these syngraft tumors displayed cell line-dependent genotypic and phenotypic differences that will make them useful preclinical models to identify and test various treatment approaches, including immunotherapies.

## 2. Results

### 2.1. Generation of Murine Glioma Cell Lines by Repeated In Vivo Passaging of Pten/p53 Deleted Cells

Recently, we described a genetic model in which tamoxifen-induced neural stem cell (NSC)-specific deletion of *Pten* and *p53* results in the development of brain tumors, which were classified as high-grade gliomas (glioblastomas) according to histopathological (necrosis and microvascular proliferation) and molecular features. In line with this classification, the tumors are positive for established glioma markers such as Gfap and Olig2 and show an intense staining for the proliferation marker Ki67. However, this genetic model (hereafter referred to as double knock-out, DKO) exhibits prolonged latency (10 to 24 months after tamoxifen injection) and incomplete penetrance of tumor development (65%) [8].

Tumor cells isolated from similar models have been successfully transplanted leading to GBM retaining characteristics of the genetic model [9,10], prompting us to generate a set of murine glioblastoma cells derived from serial in vivo passaging that would be more amenable for experimental work.

We started out with cell lines from DKO mice at two different time points of tumor development:(1)From isolated NSCs, 2 weeks after tamoxifen-induced gene deletion (transformed NSC 0; tNSC0). At this time point, DKO mice did not show any overt tumor lesions (Figure 1A); however, they all presented with an expansion of the rostral migratory stream (RMS), formed by NSCs that migrate from the sub-ventricular zone (SVZ) of the lateral ventricle (LV) to the olfactory bulb [8].(2)From isolated tumor cells of a fully established invasive high grade glioma (murine glioblastoma 0; mGB0), which occurred 12 months after the initial NSCs expansion (Figure 1A) [8].

These mGB0 and tNSC0 cells were serially transplanted into immunocompetent C57/Bl6N mice for a further two (mGB1, mGB2) and three (tNSC1; tNSC2, tNSC3) in vivo passages, respectively (Figure 1B–E). Upon each re-isolation, the tumor cells were propagated for a few passages in serum-free medium with growth factors and N2 supplement. It has been shown that these culturing conditions, favoring cell growth under spheroid conditions, avoid serum-induced cell differentiation and keep the tumor cells more similar to the primary tumors both at a phenotypic and genomic level [11].

At each in vivo passage, both tNSC and mGB cell lines gave rise to malignant and highly invasive brain tumors (Figure 1B–E) with 100% penetrance and a progressive shortening of median survival (Figure 1F,G). Remarkably, tNSC0 and mGB0 cells formed glioblastomas with similar latencies and resulted in the death of the animals within 300 days post cell transplantation. This suggests that tNSC0 cells, although isolated from DKO mice at a pre-malignant stage, already have a tumorigenic potential comparable to the mGB0 cells isolated from the fully developed tumors.

All the established glioblastoma cell lines (tNSC0, 1, 2 and 3 and mGB0, 1 and 2) were propagated in vitro for no longer than 10 passages before implantation. To ascertain that these cells retain their tumorigenicity after longer-term cultivation, tNSC3 cells were kept in serum-free culture for 60 passages and subsequently injected orthotopically into immunocompetent mice. The survival time of these animals was comparable to that of mice injected with early-passage tNSC3 cells, (Appendix A). In both cases, tumor penetrance was 100%. Moreover, tNSC3 cells injected at late passages were still able to give rise to invasive high grade gliomas (Appendix A). These results imply that the tumorigenic potential of our glioblastoma model is not affected by prolonged in vitro culture.

In summary, we showed that NSC-specific *Pten* and *p53* deletion followed by in vivo passaging of either NSCs or tumor cells via orthotopic transplantation, resulted in the development of high-penetrance brain tumors within a reasonable time frame.

### 2.2. The Murine Glioma Cell Lines Reflect the Transcriptome Heterogeneity of Human Glioblastoma

We next used RNA sequencing (RNAseq) to characterize the gene expression profiles of the cultured tNSC (tNSC0-3) and mGB (mGB0-2) cells, along with control non-transformed NSCs (ctrlNSCs, isolated from oil-injected control mice).

In order to confirm the genetic background of our tumor cells, we checked for RNAseq reads mapping to the regions of the *pten* and *p53* genes (Appendix A). As expected, we could not detect any reads corresponding to the p53 exons 2–10 or *Pten* exon 5 in any of our cell lines (Appendix A), confirming that they are derived from *Pten/p53* double knock-out NSCs. In line with these results, Gene Set Enrichment Analysis (GSEA) analysis of tNSC and mGB cells in comparison to control NSCs suggests a deregulation of p53 pathway-related genes (Appendix A). Furthermore, we called single nucleotide variants to confirm that mutations of the IDH1 and IDH2 genes, which are mutated in about 10% of human high-grade gliomas, had not spontaneously occurred during in vivo passaging and tumor development (Appendix A).

Next, we performed multidimensional scaling analysis (MDS) using all genes (Figure 2A). The analysis revealed that tNSC0 and mGB0, which were isolated from the genetic DKO model, still are closely related to the non-transformed NSCs and to each other, even though they were isolated at the beginning and end of tumor development, respectively. With in-vivo passaging, the gene expression profiles diverged, with tNSC- and mGB-derived cell lines appearing to form separate clusters (Figure 2A).

We then checked how stem cell genes change across the in vivo passages. We found that an activated NSC gene signature [12] is more highly expressed in ctrlNSC and in cancer cells at early in vivo passages, namely tNSC0 and mGB0 (Appendix A). This reflects the transcriptome-wide MDS analysis and is in line with the NSC origin of the genetic mouse model from which the cells were isolated.

It is commonly accepted that, based on gene expression signatures, IDHwt human glioblastomas can be sub-classified into three distinct molecular subtypes: classical, proneural, and mesenchymal [4]. We therefore performed single-sample (ss)GSEA with the expression signatures described in Wang et al. [4], to relate our cell lines to these well-studied human subtypes (Figure 2B). As expected, control NSCs are enriched for the classical signature, due to the large number of stemness-related genes in this set [4]. Consistent with the transcriptome-wide MDS analysis, the mGB0 and tNSC0 cell lines showed similar enrichment patterns to and clustered with control NSCs. The mGB1 cell line showed an increased enrichment score for the proneural signature, while the cell lines (tNSC2, tNSC3 and mGB2) were relatively enriched for the mesenchymal signature (Figure 2B). In vivo, these cell lines showed a progressively aggressive phenotype and decreased survival time of the recipient mice (Figure 1F,G), which is consistent with human glioblastoma, where tumors of the mesenchymal subtype have the poorest prognosis [4,13,14].

We next examined the relative expression of the individual subtype signature genes in greater detail (Figure 2C). Clustering of the cell lines using these genes recapitulated the MDS analysis on all genes (Figure 2A), with the ctrlNSCs and early passage samples (tNSC0, mGB0) grouping together and later passages forming distinct clusters based on their lineage. Altogether these results show that the neural stem-cells-derived glioblastoma cell lines serially passaged in vivo reflect, at the transcriptomic level, the heterogeneity observed in human glioblastoma.

### 2.3. Characterization of the Newly Established Syngeneic Glioblastoma Cell Lines at the Genomic Level

In order to characterize the glioblastoma cell lines at a genomic level, we performed whole exome sequencing (WES) and then analyzed copy number aberrations (CNAs) (Appendix A).

The tNSC0 and mGB0 cell lines showed a limited degree of gross chromosome instability, with the inferred copy number state still being diploid. However, copy number gains and losses acquired after the first in vivo passage were maintained in later passages (tNSC1, 2, 3 and mGB1, 2 respectively) (Figure 3A,B). In particular, the cell lines of tNSC1-3 showed gross chromosomal alterations such as the gain of a large portion of chromosome 15 and the loss of the remaining part of it, the loss of most of chromosome 14, and the gain of chromosome 6.

Overall, copy number losses were more prevalent than gains. In the tNSC cell lines, for instance, large parts of chromosomes 4, 5, 7, 16, and 18 were lost, and chromosomes 8, 14, 15, and 17 showed focal losses. This result is in line with previous reports on human glioblastoma and other tumor entities where genetic losses are more common than gains [3,15].

In order to relate observed CNAs to known human glioblastoma alterations, we used the Log Fold Change (logFC) values of genes identified via CNA calling to run a pre-ranked GSEA analysis against the human C1 positional gene set MSigDB collection. This analysis allowed us to identify murine chromosomal regions with partial deletions or amplifications that correspond to chromosomal alterations seen in human glioma samples.

In the tNSC1-3 cell lines, a large portion of the gain of the chromosome 15 is syntenic with human chromosome 8. It was recently reported that cell populations that are polyploid for this chromosome are present in human glioblastoma, and that this copy number gain can be used to identify circulating glioblastoma cells [16]. In the same cell lines, a large portion of the mouse chromosome 6 was also gained. This region corresponds to human chromosome 7 and contains an amplification of the Met receptor with a consistently high copy number logFC value (0.56, 1.16 and 1.31 for tNSC1, tNSC2 and tNSC3, respectively). MET is commonly overexpressed or amplified in human glioblastoma. However, in our cell lines, the RNA seq data showed that this amplification did not affect the expression levels of Met, suggesting that further genetic and/or epigenetic events might affect Met expression in accordance with what was also described in human GBM [17,18]. The tNSC1-3 cell lines also showed a notable deletion in a major part of chromosome 14 including the mouse Rb1 gene that has been reported to be lost or mutated in a subset of human glioblastoma patients [19]. According to the CNAs, we detected a partial loss of the mouse Rb1 gene in these cell lines, resulting in a significant (up to 1.6logFC) reduction in Rb1 RNA expression in the tNSC cell lines compared to control NSCs. Furthermore, GSEA performed on the expression data shows a highly significant enrichment of Rb1 regulated genes that were overexpressed in tNSC3 cells in comparison to the control NSCs (Appendix A). As this gene set represents genes typically downregulated by Rb1 [20], these data suggest that the chr14 deletion contributes to the loss of the tumor-suppressive function of Rb1.

The mGB cell lines showed a smaller number of gross genomic alterations in comparison to the tNSCs. Performing the same pre-ranked GSEA analysis as previously described, we noted a single-copy gain of mouse chromosome 10 in the mGB0 and mGB2 cell lines. This region of the mouse chromosome corresponds to the human 12q13 region, which has been previously reported to be amplified in glioblastoma [21]. The Erbb3, Gli1 and Mdm2 genes are all present in this region, and their amplification is known to have effects in human glioblastoma [22,23,24]. The amplifications of Erbb3 and Mdm2 correlated with a 6.2 and 1.9 logFC increase in RNA expression, respectively, compared to control NSCs.

These data illustrate that our murine cell lines harbor genetic alterations known to be relevant in human glioblastomas.

### 2.4. The Tumors Derived from the Syngeneic Cell Lines Resemble Human Glioblastomas

Upon orthotopic transplantation, all the tNSC and mGB cells lead to the development of high grade gliomas. Although tumor cells were always implanted in the right hemisphere, the resulting tumors extended into the contralateral hemisphere (Figure 1B–E), underlining the invasive potential of these tumor cells. In general, tumors obtained from mGBs cells showed intratumoral hemorrhages (Figure 1C, right panel) and extensive necrotic areas (Figure 1D, right panel) in 40% and 15% of the cases, respectively. In comparison, 20% and 0% of tNSCs-derived tumors show similar features.

The glioblastoma cell lines derived from the last in vivo passages (tNSC3 and mGB2) were characterized in more detail. The cells were stably transduced with GFP to enable visualization and unambiguous identification of tumor cells in tissue samples. Immunohistochemical staining of GFP-expressing tNSC3 and mGB2 cells in formalin-fixed paraffin-embedded (FFPE) brain sections showed that the tumor margins of the tNSC3- and mGB2-derived gliomas (Figure 4A,B) were not well delineated due to tumor cell invasion of the surrounding brain parenchyma (Figure 4C,D), as is characteristic of human glioblastoma. As depicted in Appendix A, the highly invasive growth pattern is also clearly visible by MRI imaging, a feature which would allow following invasion in vivo over time.

In addition, these tumors had histopathological features of high-grade gliomas, such as microvascular proliferation and the presence of mitotic figures (Figure 4E–H). These characteristics are also observed in tumors derived from tNSC3 cells after prolonged in-vitro culture (Appendix A).

We used antibodies against the vascular endothelial markers CD31 and CD34 to detect blood vessels and observed less but bigger vessels in the tumor areas as compared to the surrounding brain parenchyma (Appendix A).

The syngeneic gliomas showed strong positivity for the proliferation marker Ki67 (Figure 4I,J), consistent with observations in human GBM specimens. The strong reactivity to Ki67 is evident also in tumors derived from earlier in-vivo-passaged cell lines (Appendix A), indicating that all xenotransplanted tumors exhibit high proliferative capacity.

tNSC3- and mGB2-derived tumors were positive for the branched-chain amino acid transaminase 1 (BCAT1) (Figure 4I,J), which was previously described to be exclusive to human gliomas with wild-type IDH1 and IDH2 [25]. Furthermore, immunohistochemical analysis of the tumors revealed prominent staining for the human glioma markers Olig2 (Figure 4K,L) and Gfap (Figure 4M,N).

Next, we examined the tumor microenvironment with a focus on immune cells. Staining of tNSC3 and mGB2 tumors for Cd11b and Iba1 indicated a strong presence of myeloid cells (Figure 4O–R). These results are in line with the observation that human glioblastomas consist of 30–50% myeloid cells, mainly tumor associated macrophages (TAMs) [26].

Consistent with the immunocompetent background of the recipient mice, we additionally observed lymphocyte infiltration as detected by immunofluorescence staining for CD3, CD4 and CD8, with CD4 positive cells being the most abundant lymphocyte population (Figure 5A–C). Of note, the lymphocytes are localized within the tumor core as well as in the invasive front, as indicated by CD4 positive lymphocytes in the corpus callosum, where GFP positive cells migrate to the contralateral hemisphere (Figure 5D).

Altogether, these data show that the tumors formed after transplantation of the newly generated mouse cell lines resemble human glioblastoma with respect to histopathological and molecular features and to immune cell infiltration.

## 3. Discussion

The subventricular zone (SVZ) of the adult human brain contains a population of neural stem cells (NSCs), which are capable of multilineage differentiation [27,28]. Mutations of key tumor suppressors in SVZ cells have been shown to initiate the development of glioblastomas of diverse phenotypes [29,30]. We previously described a genetic mouse model with Tlx [31] promoter-driven NSC-specific knockout of *Pten* and *p53* genes [8], which recapitulates prominent features of human glioblastoma. This is in accordance with similar GEMM models where inactivation of the same tumor suppressor genes was achieved in GFAP positive cells [9,10,17].

Tumor cells isolated from some of these models have been successfully transplanted leading to GBM retaining characteristics of the genetic model [9,10]. While Jacques and colleagues transformed the NSCs in vitro prior to transplantation into immunocompetent mice, Zheng and colleagues reinjected tumor cells using immunocompromised SCID mice as recipients.

In contrast, we isolated cells from our GBM model at different stages of tumor progression and transplanted them serially into fully immunocompetent C57/Bl6 mice. With this approach, we were able to generate cell lines that, after syngeneic transplantation, reproducibly formed high-grade gliomas, displaying key characteristics of human glioblastoma. One of the most striking and clinically relevant features of human glioblastoma is their highly invasive growth. Glioblastoma cells typically infiltrate large regions of the brain including the brain parenchyma, the corpus callosum and the contralateral hemisphere. This invasive growth constitutes one of the biggest clinical challenges, preventing complete surgical resection and almost invariably resulting in tumor recurrence. The mouse glioblastomas resulting from the transplantation of our cell lines display remarkably similar infiltrative growth patterns with a spread to the contralateral hemisphere in all cases. Furthermore, they showed necrotic areas and microvascular proliferation, both defining characteristics of human glioblastoma. In contrast, the prevalently utilized syngeneic mouse glioblastoma model, the cell line GL261, is lacking most of these features and presents with bulky tumor growth and no obvious infiltration of the adjacent normal brain tissue [7].

Transcriptome analysis of the cells revealed cell line-specific enrichment of RNA expression signatures of the classical, proneural and mesenchymal human glioblastoma subtypes. This suggests that, despite their common cell of origin, the cell lines through in vivo passaging have diversified to present a level of heterogeneity resembling that of human glioblastoma. In line with these findings, genome analysis identified copy number aberrations that were conserved through in vivo passaging, including some that correspond to known chromosome aberrations of human glioblastoma.

Closely recapitulating genomic and histopathological features of human GBM is a key prerequisite to test novel therapeutic approaches such as new treatment protocols and/or specific drug candidates [32].

Consistently, in the last years, the use of patient-derived organoid and xenograft models has increased drastically. Recently, patient-derived organoid models have been described as useful tools to study the genetic background and intratumoral heterogeneity of human GBM cells [33]. However, the requirement to use immuno-compromised mice is a major constraint of all patient-derived in vivo models, in particular with respect to studying the interaction between tumor cells and cells of the immune microenvironment. The development of new treatment options aiming at modulating the immune response e.g., with modulatory drugs such as immune checkpoint blockers, or to exploit immune cells as therapeutic vehicles, such as chimeric antigen receptor (CAR)-T cells, is largely dependent on models having a functional immune system. Our mouse model fills this gap by providing the possibility to investigate immune-modulatory therapies in vivo in a murine context, recapitulating features of human GBM.

Additionally, our transplantation models have been recently used to study the role of the SOX10 master regulator transcription factor in the determination of glioblastoma transcriptional and phenotypic transitions [34]. In this study, the mGB1 syngeneic, orthotopic transplantation model was used to show that, upon suppression of Sox10, syngraft tumors not only acquired a more aggressive growth phenotype, resulting in significantly worse survival of the animals, but also showed a much greater extent of infiltration by myeloid cells. These results demonstrate that our panel of cell lines can successfully be utilized in deciphering determinants of glioblastoma malignancy and, thus, contributes to future efforts in targeting this disease.

## 4. Materials and Methods

### 4.1. Animals

Tlx-CreERT2/*p53*-floxed/*Pten*-floxed (double knockout [DKO]) mice were described in [8]. To induce recombination of floxed alleles, 4-week-old mice were injected intraperitoneally with 1 mg tamoxifen (S5007 Sigma) in 5% ethanol and 95% oil (T5648 Sigma) for 5 consecutive days. Animal experiments were approved by (i) the German responsible authority (Regierungspräsidium Karlsruhe) and performed in conformity with the German law for Animal Protection (animal license number: G-156/15, G-199/11) and (ii) the Animal Care and Use Committee of the San Raffaele Hospital (IACUC no. 979) and authorized by the Italian Ministry of Health and local authorities accordingly to Italian law.

### 4.2. Cell Isolation

NSCs were isolated from the subventricular zone (SVZ) of mice 2 weeks after the injection of either tamoxifen (tNSC0) or oil (ctrlNSCs) according to [35]. Glioma-bearing mice displaying termination criteria such as loss of >20% body weight, neurological deficits or poor general condition were euthanized with carbon dioxide; brains were minced and dissociated in Leibovitz-L15 (Life Technologies, Pittsburgh, PA, USA) containing 10 U/mL papain, 5 mM EDTA, and 200 U/mL DNAse. Cells were grown according to [35].

### 4.3. Cell Transduction

Lentiviral transduction with a construct encoding eGFP (Plasmid #14883, Addgene, Watertown, MA, USA) was performed in order to label the cells. For virus production, one 10-cm dish of HEK293T cells was transfected with 8 μg target vector; 4 μg psPAX2; 2 μg pVSVg; and 42 μg polyethylenimine (Alfa Aesar, Haverhill, MA, USA). HEK293T cells were cultivated in N2-supplemented serum-free medium. Virus-containing medium was transferred from HEK293T cells to the target cells and replaced by cultivation medium after 24 h.

### 4.4. Orthotopic Intracranial Injections

Mice were anesthetized with isoflurane and placed on a stereotaxic frame. A total of 5 × 10^5^ cells in 2 µL phosphate-buffered saline were injected 2 mm lateral (right) to the bregma and 3 mm deep at a flow of 0.2 µL/min using a 10-µL precision microsyringe (World Precision Instruments, Inc, Sarasota, FL, USA) with a 34G needle.

### 4.5. Magnetic Resonance Imaging

Magnetic resonance imaging (MRI) was performed on a 7.0 Tesla preclinical scanner (Bruker Biospin, Germany), equipped with a specific surface coil for mouse head. All mice were anaesthetized with isoflurane (3% for induction and 2% for maintenance), in a 95–98% O_2_ mixture. During acquisition, mice were positioned prone on a dedicated temperature control bed to prevent hypothermia. Coronal images of the entire brain (14 slices of 0.75 mm thickness) were acquired using a Turbo Spin Echo T2 sequence (TR = 3000 ms; TE = 48 ms; turbo-factor = 10; pixel-size 112 μm × 88 μm). MRI imaging was performed 41 days after xenotransplantation.

### 4.6. Survival Analysis

Length of animal survival was measured by means of Kaplan–Meier estimate using nonparametric log-rank tests.

### 4.7. Immunohistochemistry and Immunofluorescence Staining

Formalin fixed paraffin-embedded sections (5 µm) were stained as described previously [36] using the following primary antibodies: rabbit anti-Ki67 (ab15580, Abcam, Cambridge, England), rabbit anti-OLIG2 (ab109186, Abcam), rabbit anti CD11b (ab133357, Abcam, Cambridge, England), rabbit anti Iba1 (019-1974, Wako, Japan), chicken anti GFP (ab1397, Abcam, Cambridge, England), mouse anti Bcat1 (TA504360, OriGene), mouse anti-GFAP (644701, BioLegend, San Diego, CA, USA), rabbit anti-CD31 antibody (ab28364, Abcam, Cambridge, England), and rat anti-CD34 (NB600-1071, Novus Biologicals, Centennial, CO, USA).

For CD3, CD4 and CD8 stainings, 4% PFA fixed OCT-embedded brains were used. In particular, 16 µm sections were stained as described [37] using rabbit anti CD3 (ab16669, Biolegend, San Diego, CA, USA), rat anti CD4 (100505, Biolegend, San Diego, CA, USA), and rabbit anti CD8 (EPR20305, Abcam, Cambridge, England).

Secondary antibodies were: goat anti-rabbit Alexa Fluor 546 (A11071 Thermo Fischer Scientific, Pittsburgh, PA, USA), goat anti-mouse Alexa Fluor 555 (A21422, Thermo Fischer Scientific, Pittsburgh, PA, USA), goat anti-rabbit Alexa Fluor 488 (A11008 Thermo Fischer Scientific, Pittsburgh, PA, USA), and goat anti-rat Alexa Fluor 546 (A11081, Thermo Fischer Scientific, Pittsburgh, PA, USA). For immunohistochemistry, the biotinylated goat anti-rabbit immunoglobulin-G (BA1000) was used together with DAB peroxidase (horseradish peroxidase) substrate kit (SK-4100), both from Vector Laboratories, Burlingame, CA, USA.

### 4.8. Image Acquisition

Pictures were captured with Zeiss Axio-Scan.Z1 using ZEN software (Zeiss, Oberkochen, Germany) or with the MEA53100 Eclipse Ti-E inverted microscope (Nikon, Japan) using MQS31000 NIS-ELEMENTS AR software (Nikon, Japan) with camera MQA11550 DS-Qi1MC for bright field images and MQA11010 DS-Fi1 for immunofluorescent images.

CD3-, CD4- and CD8-stained sections were imaged using Leica SP8 confocal microscopy with 20× objective. Alignment of images to obtain the whole brain sections was done applying the automatic stitching of images using the Leica dedicated application (Las-X, Leica, Wetzlar, Germany).

### 4.9. Library Preparation and Target Enrichment of Genomic DNA for Whole Exome DNA Sequencing

Genomic DNA was fragmented to a size range of 150–250 bp using the Covaris S220 instrument. The fragmented DNA was end repaired and adenylated. The SureSelect Adaptor was then ligated followed by a pre-amplification. After library preparation, target regions were captured by hybridization of biotinylated baits (library probes) using Agilent SureSelect XT Human All Exon V6 (Agilent, Santa Clara, CA, USA). Captured target sequences were then isolated using streptavidin-coated magnetic beads. Subsequently, the appropriate 8-bp single index tags were added during sequencing library amplification. All steps were done using Agilent SureSelect XT Reagent Kit (Agilent, Santa Clara, CA, USA). Final quality control was done using Qubit 3 fluorometer and Agilent Bioanalyzer 2100 (Agilent, Santa Clara, CA, USA). The libraries were sequenced in paired-end mode (2 × 50 nt) on a NovaSeq6000 S2 Flowcell (Illumina, San Diego, CA, USA) resulting in ~200 million distinct sequencing reads per library.

### 4.10. mRNA-Focused Library Preparation of Total RNA for NGS

Poly(A)-positive mRNA transcripts were isolated from the total RNA by binding to magnetic oligo(d)T beads (RNA purification beads). After elution from the beads and fragmentation, the mRNA was reverse-transcribed during first and second strand synthesis, yielding double-stranded cDNA. The ends of the cDNA molecules were end-repaired, and an adenosine overhang was added to all 3′ ends (A-tailing step) to facilitate the ligation of the adapter molecules. These adapters have individual dedicated index sequences and add the Illumina-specific sequences that are needed for amplification, flow cell hybridization and sequencing. Eight beats-per-minute single-index NEXTflex DNA barcodes (PerkinElmer, Waltham, MA, USA) were used. The final ligation product was amplified via PCR. Double-stranded cDNA sequencing libraries were further checked for quality and quantity using Qubit 3 fluorometer and Agilent Bioanalyzer 2100. The libraries were sequenced in paired-end mode (2 × 50 nt) on a NovaSeq6000 S2 Flowcell resulting in ~50 million distinct sequencing reads per library.

### 4.11. Bioinformatic Data Analysis

#### 4.11.1. WES Data—Basic Processing

Whole exome sequencing paired end reads in fastq format were aligned using BWA (v0.7.15) to the mm10 reference genome with options: -M -T 0. Duplicates were marked using sambamba (0.6.6).

#### 4.11.2. WES—CNV Analysis

CNV analysis was performed using CNVkit (v0.9.6) [38]. Tumor WES samples (tNSC0-3, GBM0-2) were analyzed with reference to the two normal WES samples, ctrlNSC and normal splenocytes. The mm10 genome. fasta was used to calculate accessibility as follows: access mm10.fa -s 10000 –o mm10_accessibility.bed. The S0276129 bedfile with Agilent capture array loci, lifted over to the mm10 genome, was used as the target regions. Regions were annotated using UCSC’s mm10 flat reference [39].

Calling was performed with the batch command and the following options: batch TUMOR_SAMPLES --normal NORMAL_SAMPLES --targets S0276129_mm10.bed --fasta mm10.fa --access mm10_accessibility.bed --annotate refFlat.txt --drop-low-coverage. Copy number state segmentation was performed using the ‘cbs’ method as implemented in the R package DNAcopy (v1.54.0).

The results of the CNV calling were visualized using the R package circlize (v0.4.4) [40] in R v3.5.1 (Vienna, Austria). Regions overlapping annotated segmental duplications (downloaded from the UCSC Table Browser, accessed 25 January 2020) were removed before plotting due to potential issues with inferring copy number states in these repetitive regions. For ease of visualization, extreme values with CNVkit log2 copy ratio <−2 or >2 were set to this minimum and maximum. The integer CNVkit segmented copy number states were smoothed using the rollarray command from zoo (v1.8-2) with the options width = 301, partial = TRUE.

The identification of human syntenic regions with observable CNAs in the tNSC and mGB cell lines was done using GSEA against the human “c1.all.v7.0.symbol” dataset with gene names adapted to mouse. The analysis was performed as a pre-ranked test using logFC CNA values produced in the CNA calling.

#### 4.11.3. RNA-Seq Data—Basic Processing

RNA-seq reads in. fastq format were aligned with STAR (v2.5.2a) [41] to the mm10 reference genome and the Gencode M2 reference transcriptome, using the following options: --outFilterMismatchNmax 2 --outFilterMismatchNoverLmax 0.05 --alignIntronMax 1 --outFilterMatchNminOverLread 0.95 --outFilterScoreMinOverLread 0.95 --outFilterMultimapNmax 10 --outFilterIntronMotifs RemoveNoncanonical --outFilterType BySJout --outSAMunmapped Within --outSAMattributes Standard --alignIntronMin 21 --outFilterMatchNmin 16. Overlapping read pairs were clipped using the clipOverlap tool in bamUtil (v1.0.9) [42]. The raw counts matrix was generated using featureCounts (v1.5.3) [43] using options -Q 255 -p -t exon against the M2 transcriptome. Aligned data were quality controlled using RSeQC (v2.6.4) [44].

#### 4.11.4. RNA-Seq—Transcriptome Analysis

This analysis was performed in R (v3.4.3). The raw counts matrix produced by featureCounts was pre-filtered to remove genes with low expression, keeping those with >10 reads in >2 samples, and normalized using the limma-voom (v3.34.4) function voomWithQualityWeights. Gencode M2 Ensembl Gene IDs were mapped to other identifiers using org.Mm.eg.db (v3.5.0). Mouse gene identifiers were mapped to their human homologues using the Jackson Laboratory Mouse-Human HomoloGene table [45].

Gene Set Enrichment Analysis was performed for each condition against the Wang 2017 subtype classifier [4]. Conditions with multiple replicates (ctrlNSC and tNSC3) were averaged, and then each was tested against the 50 subtype genes from the Wang classification, mapped to mouse homologues as described above, using the ssgsea option in GSVA (v1.26.0) [46]. MDS plots were generated using the complete expression matrix, with replicates averaged as above.

Heatmaps were visualized using ComplexHeatmap (v1.17.1) [47] and other graphics using ggplot2 (v2.2.1). The rows/columns in both heatmaps in Figure 3 were clustered with the Ward.D2 method using 1-Pearson’s correlation as the distance metric.

#### 4.11.5. RNA-Seq—Stemness Signature Analysis

The activated NSC signature is described in Table S6 of [12]. Of the 243 unique genes in the signature, 164 were present in the analyzed gene expression matrix.

#### 4.11.6. WES/RNA-Seq—Idh1/2 Hotspot Mutation Analysis

Single nucleotide variants (SNVs) were called in the WES and RNA-seq data using BCFTools (v1.9) [48] at the three hotspots (Idh1: R132 = 1:65170977-65170979; Idh2: R140 = 7:80099112-80099114; R172 = 7:80099016-80099018). bcftools mpileup was used on the full set of samples for each data type, and consensus calling performed with bcftools call –c.

The GSEA for the enrichment in the p53 and rb1 transcriptional signatures was performed as a pre-ranked analysis of the genes differentially expressed between tNSC and mGB cell lines and the ctrlNSCs. Only genes with FDR < 0.01 were considered for the analysis. The expression signatures used were MSigDb “h.all.v7.0.symbols” and “c2.all.v7.0.symbols” adapted for mouse gene names, respectively.

#### 4.11.7. Code Availability

Scripts used to process and analyse the WES and RNAseq data are available at (Analysis code repository, https://github.com/mncfletcher/angel-gb-gemms, 2019).

### 4.12. Data Availability

RNA seq and WES data have been deposited at the GEO; accession number GSE145559.

## 5. Conclusions

In summary, our set of GBM cell lines presents the following benefits:(i)They can be re-transplanted into fully immunocompetent C57/Bl6 mice, which allows the use of a broad variety of already available knock-out, knock-in mice as recipients e.g., to identify tumor-extrinsic factors affecting tumor growth and/or invasion.(ii)They allow testing of immunomodulatory therapies.(iii)They are derived from serial transplantations of cells isolated in vivo at different stages of tumor development and exhibit different latencies. This allows long-term treatment options with wide therapeutic windows, not possible with many current syngeneic animal models.(iv)They are characterized both at transcriptomic and genomic level. This will enable researchers to choose a cellular model based on the molecular subtype, extract information about their genes of interest and use the appropriate cell line for their studies. Moreover, additional genetic modifications of the cell lines in vitro, i.e., by CRISPR/Cas9 or viral transduction, are easily possible.(v)They allow following tumor evolution by tracking retention and acquisition of molecular alterations over serial in vivo passaging, thus reflecting the range of genotypic and phenotypic heterogeneity that is also observed in human glioblastoma.(vi)They are available to be shared with the scientific community.

Altogether, these assets make our GBM cell lines an attractive, unique and versatile tool for basic and translational glioblastoma research.

## Figures and Tables

**Figure 1 cancers-13-00230-f001:**
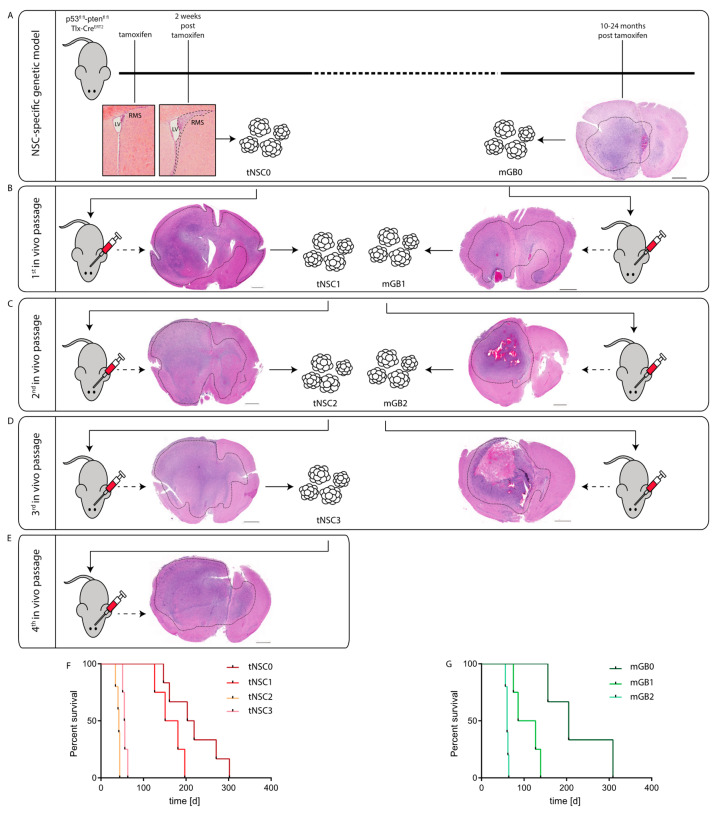
Generation of syngeneic glioma cell lines through in-vivo passaging of *Pten*/*p53*-deleted cells isolated from a genetic glioma model. (**A**) Schematic representation of the *Pten*/*p53* genetic glioma model from which NSCs have been isolated either at early pre-malignant stage (left panel) or from a full-blown tumor (right panel). Scale bar = 1000 µm. Left panel: lateral ventricle (LV) with rostral migratory stream (RMS) at time of tamoxifen injection and at 2 weeks after tamoxifen-induced *Pten/p53* recombination. Sections were stained with hematoxylin and eosin (H&E). Right panel: section of a full-blown tumor developed 18 months after tamoxifen-induced *Pten/p53* deletion. Section was stained with hematoxylin and eosin (H&E). Scale bar = 1000 µm Dotted line shows tumor area. (**B**) First in vivo passage of tNSC0 (left panel) and mGB0 (right panel). Representative picture of a tNSC0-derived (left panel) and mGB0-derived (right panel) glioma. Sections were stained with hematoxylin and eosin (H&E). Scale bars = 1000 µm, (**C**) second in vivo passage of tNSC1 (left panel) and mGB1 (right panel). Representative picture of a tNSC1-derived (left panel) and mGB1-derived (right panel) glioma. Sections were stained with hematoxylin and eosin (H&E). Scale bars = 1000 µm. (**D**) Third in vivo passage of tNSC2 (left panel) and mGB2 (right panel). Representative picture of a tNSC2-derived (left panel) and mGB2-derived (right panel) glioma. Sections were stained with hematoxylin and eosin (H&E). Scale bar = 1000 µm, (**E**) fourth in vivo passage of tNSC3 (left panel). Representative picture of a tNSC3-derived (left panel). Sections were stained with hematoxylin and eosin (H&E). Scale bar = 1000 µm. (**F**) Kaplan–Meier survival curve of mice transplanted with tNSC0, tNSC1, tNSC2, and tNSC3 glioma cell lines. Time = days. tNSC0 (*n* = 7), tNSC1 (*n* = 6), tNSC2 (*n* = 6), tNSC3 (*n* = 7). (**G**) Kaplan–Meier survival curve of mice transplanted with mGB0, mGB1, and mGB2 glioma cell lines. Time = days. mGB0 (*n* = 6), mGB1 (*n* = 6), mGB2 (*n* = 6).

**Figure 2 cancers-13-00230-f002:**
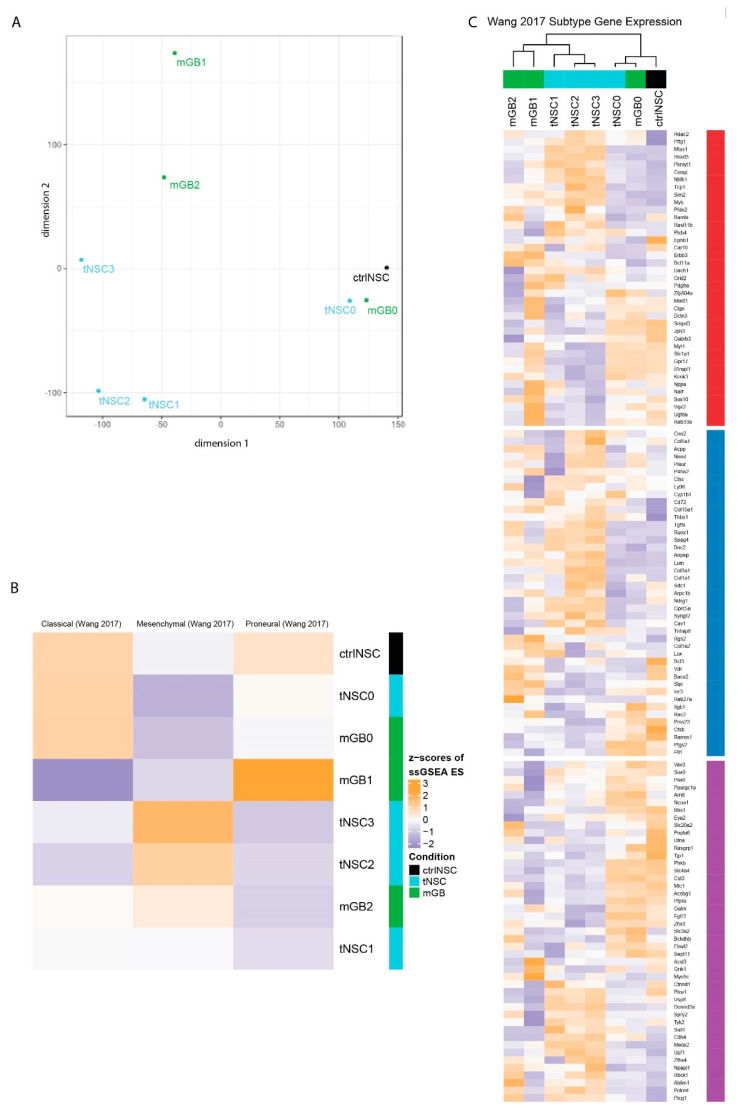
Transcriptomic characterization of the new glioma cell lines finds distinct expression profiles that correspond to known human glioblastoma subtypes. (**A**) Multidimensional scaling (MDS) analysis of whole transcriptomes from the newly generated glioma cell lines (tNSC0-3, mGB0-2) and of non-transformed NSCs (ctrlNSCs). (**B**) Heatmap showing z-scores of ssGSEA enrichment scores of the murine glioma cell lines for published glioblastoma subtype gene expression signatures. (**C**) Heatmap of z-scores of expression of glioblastoma subtype genes, as defined in Wang et al. Genes are labeled with their subtype. The gene expression of cell lines with sample replicates (tNSC4 and ctrlNSC: both *n* = 3) was averaged.

**Figure 3 cancers-13-00230-f003:**
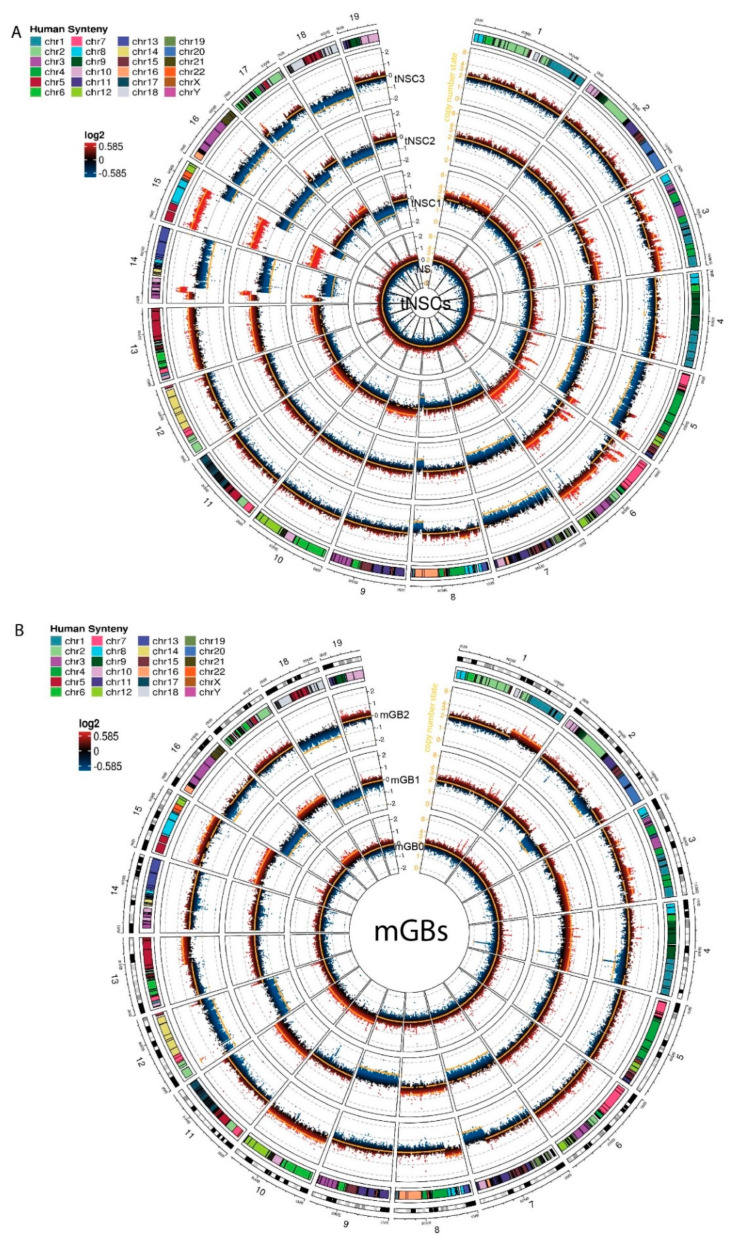
Copy number aberration analysis identifies specific chromosomal aberrations of the new murine glioma cell lines. (**A**,**B**) Circos plots showing copy number aberrations (CNA) of tNSCs (**A**) and mGBs (**B**) cell lines. CNVkit-calculated log2 ratios for each genomic bin are plotted as colored points (black y-axis scale; red corresponds to increased log2 ratios, and blue to decreased), with the inferred copy number state plotted as an orange line (orange y-axis scale). Each circular sector represents one chromosome. Human syntenic regions are shown at the top of each chromosome.

**Figure 4 cancers-13-00230-f004:**
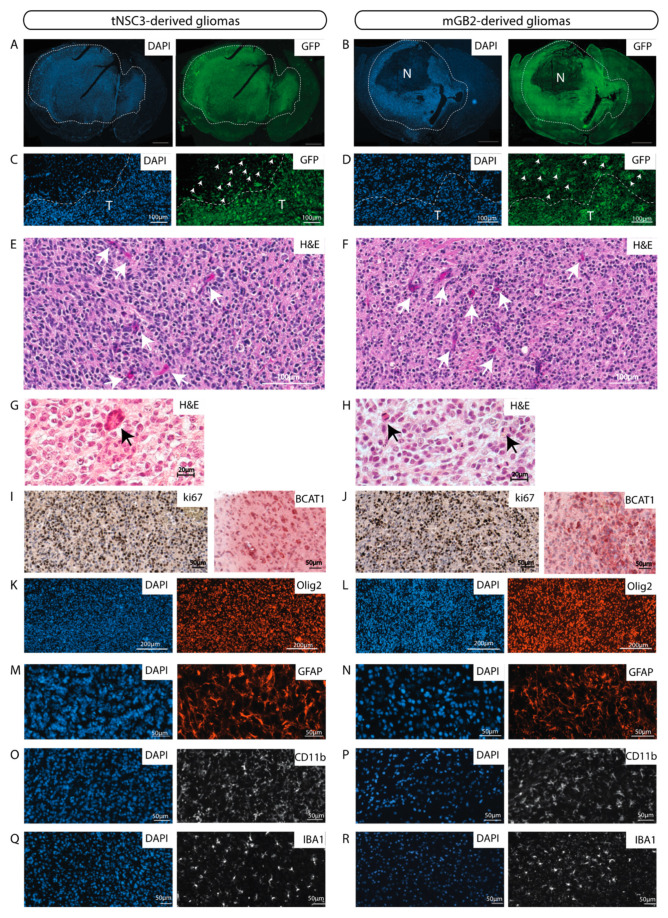
The murine GBM cell lines give rise to orthotopic tumors with characteristics of human glioblastomas. (**A**,**B**) GFP-Immunofluorescence staining (green) of tNSC3 (**A**) and mGB2 (**B**) orthotopic tumors. Cellular nuclei are stained with DAPI and pseudocolored in blue. Scale bars, 1000 µm. Dotted lines show tumor area. N indicates a necrotic area in the mGB2 orthotopic glioma. (**C**,**D**) Zoom-in of images from A and B, respectively, showing an area at the tumor border. T indicates tumor area delineated by dotted lines. Arrows denote GFP-positive glioma cells invading the surrounding normal brain parenchyma. (**E**–**H**) Histopathological features of orthotopic tNSC3 (**E**,**G**) and mGB2 (**F**,**H**) gliomas. Sections were stained with hematoxylin and eosin (H&E). Arrows in **E**, **F** denote areas of microvascular proliferation; arrows in **G**, **H** indicate mitotic figures. (**I**,**J**) Ki67 expression (left) and BCAT1 (right) in orthotopic tNSC3 (**I**) and mGB2 (**J**) gliomas detected by immunohistochemistry. The sections were counterstained with hematoxylin. (**K**–**R**) Immunofluorescence staining for Olig2 (**K**,**L**), Gfap (**M**,**N**), Cd11b (**O**,**P**), and Iba1 (**Q**,**R**) within tumor areas from orthotopic tNSC3 (**K**,**M**,**O**,**Q**) and mGB2 (**L**,**N**,**P**,**R**) gliomas. Cellular nuclei are stained with DAPI and pseudocolored in blue.

**Figure 5 cancers-13-00230-f005:**
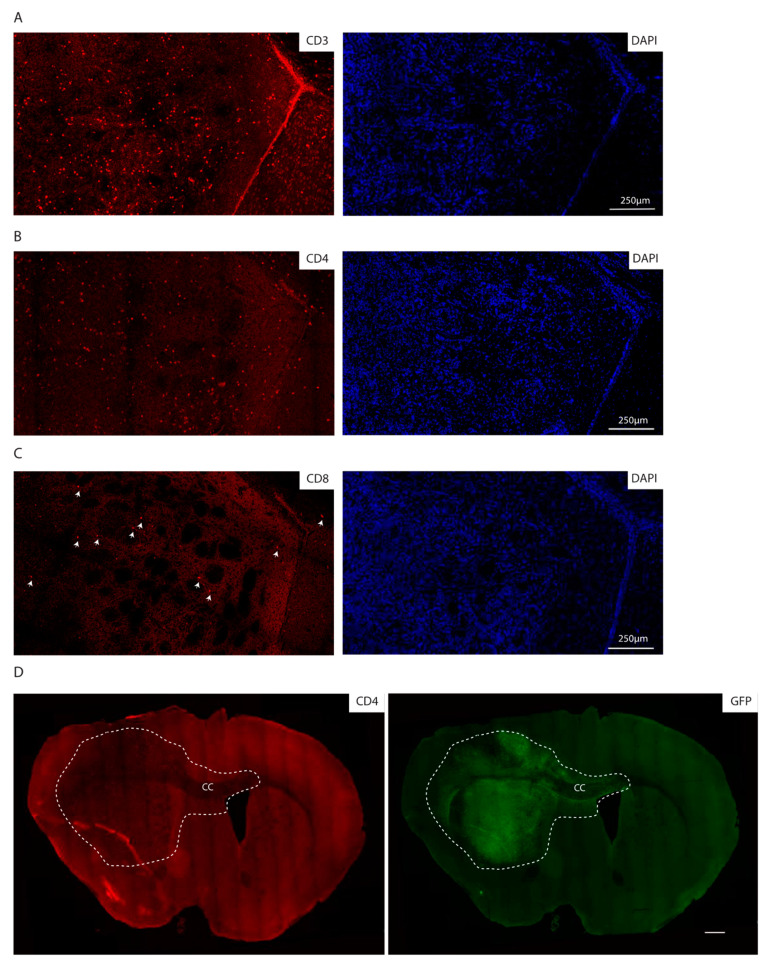
The syngeneic GBM show lymphocyte infiltration. (**A**–**C**) Immunofluorescence staining of the core of an mGB2 orthotopic tumor for CD3 (**A**), CD4 (**B**), CD8 (**C**). Cellular nuclei are stained with DAPI and pseudocolored in blue. Arrows indicate CD8 positive cells. (**D**) Immunofluorescence staining of an mGB2 orthotopic tumor (entire brain section) for CD4 (in red) and GFP (in green). Scale bar, 500 µm. Dotted lines show tumor area. CC indicates corpus callosum.

## Data Availability

The genomic data presented and analysed in this study are openly available at the GEO at GSE145559.

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
