# Peer review of "A Set of Cell Lines Derived from a Genetic Murine Glioblastoma Model Recapitulates Molecular and Morphological Characteristics of Human Tumors"

_cancers, 2021, doi:10.3390/cancers13020230_

Round 1
Reviewer 1 Report
The manuscript by Costa et al describes a very interesting and novel preclinical model for glioblastoma (GBM). Not many are the immunocompentent mouse models for GBM, such as GL261 and CT2A. Anyway, with the advent of immunotherapy, the need of syngeneic models capable to represent different subtypes of GBM has become very urgent and this manuscript is very timely. The clinical and preclinical relevance of this manuscript is very high.
The manuscript is clear and well written. The authors have done a number of genomic, transcriptomic and histological analysis to prove this model as suitable and useful for relevant preclinical studies.
There are only a few points that are unclear to me and that I feel should be addressed, nonetheless, I think the authors have done an excellent job presenting the model and its relevance.
Remaining concerns and open questions:
- How do you determine when to explant and reimplant the cells? When the mouse has clinical signs or at a specific tumor size? Is the time between in vivo passages always the same or it decreases after several passages?
- Which are the changes in stem cell genes happening across the in vivo passages. The expression of stem cells genes may be higher and higher across the passages, since the tumors are more aggressive with a hypothetical selection of the most aggressive clones. Is the proliferation rate (in vitro and in vivo) always the same across the passages?
- Figure 1: it is always the same picture that is used to represent the different in vitro cells.
- Figure 4: it could be nice to have the histology from earlier-passage tumors to compare their invasive capacity and proliferation. A staining for CD34 or CD31 could be useful to study microvessel density. Finally, I strongly recommend to make the figure consistent, with the same colors (the labels are not visible, please change them)
- How long can you keep the cells in culture with no morphological changes and differentiation?
- A section for statistical analyses is missing in the M&M. Figure 1 needs statistics for example.
- Description of the survival studies (and Kaplan-Meyer) presented in Figure 1 is missing in the M&M.
Author Response
The manuscript by Costa et al describes a very interesting and novel preclinical model for glioblastoma (GBM). Not many are the immunocompentent mouse models for GBM, such as GL261 and CT2A. Anyway, with the advent of immunotherapy, the need of syngeneic models capable to represent different subtypes of GBM has become very urgent and this manuscript is very timely. The clinical and preclinical relevance of this manuscript is very high.
The manuscript is clear and well written. The authors have done a number of genomic, transcriptomic and histological analysis to prove this model as suitable and useful for relevant preclinical studies.
There are only a few points that are unclear to me and that I feel should be addressed, nonetheless, I think the authors have done an excellent job presenting the model and its relevance.
Remaining concerns and open questions:
- How do you determine when to explant and reimplant the cells? When the mouse has clinical signs or at a specific tumor size? Is the time between in vivo passages always the same or it decreases after several passages?
- We explanted the tumor cells when the mice displayed clinical signs. This information is now added in the Material and Methods section, lines 419-420: “Glioma-bearing mice exhibiting termination criteria such as loss of >20% body weight, neurological deficits or poor general condition were euthanized…”
Since the survival of the animals diminished at each in vivo passage, as depicted in Figure 1 F and G, the time period between in vivo passages consequently decreased as well.
- Which are the changes in stem cell genes happening across the in vivo passages. The expression of stem cells genes may be higher and higher across the passages, since the tumors are more aggressive with a hypothetical selection of the most aggressive clones. Is the proliferation rate (in vitro and in vivo) always the same across the passages?
- In order to answer this question, we analyzed the transcriptomic data with respect to this issue. The analysis is now included in the Material and Method section, under the paragraph called “RNA-seq – stemness signature analyisis”. In the Results section, lines 201-205 we now added: “We then checked how stem cell genes changed across the in vivo passages. We found that an activated NSC gene signature (Beckervordersandforth et al., 2015) is more highly expressed in ctrlNSC and in cancer cells at early in vivo passages, namely tNSC0 and mGB0 (Figure S2E). This reflects the transcriptome-wide MDS analysis and is in line with the NSC origin of the genetic mouse model from which the cells were isolated.” Opposite to what Reviewer 1 hypothesized, the cell lines with higher expression of stemness-related genes are ctrlNSC, tNSC0 and mGB0. This, however, could be explained by the fact that the genetic model, from which we isolated the tumor cells, is NSC-specific.
In order to check proliferation in vivo, we stained the syngeneic tumors for the proliferation marker Ki67. We now added the following lines 309-312:” The syngeneic gliomas showed strong positivity for the proliferation marker Ki67 (Figure 4I and J) consistent with observations in human GBM specimens. The strong reactivity to Ki67 is evident also in tumors derived from earlier in vivo-passaged cell lines (Figure S5) indicating that all xenotransplanted tumors exhibit high proliferative capacity.”
To note, compared to the previous version of the manuscript Figure S5 is newly added and contains Ki67 staining of gliomas derived not only from the cells of the last in vivo passage but also of earlier in vivo passages.
- Figure 1: it is always the same picture that is used to represent the different in vitro cells.
- We used only one picture as a representative one depicting glioma cells grown as spheroids. However, in order to avoid confusion, we now replaced the spheroid picture with a schematic representation of spheroids. Please see Figure 1A,B,C,D
- Figure 4: it could be nice to have the histology from earlier-passage tumors to compare their invasive capacity and proliferation. A staining for CD34 or CD31 could be useful to study microvessel density. Finally, I strongly recommend to make the figure consistent, with the same colors (the labels are not visible, please change them)
- The histology of earlier-passage tumors is already included in Figure 1A,B,C,D,E while for the proliferation we now added the Figure S5 showing Ki67 staining of earlier-passage tumors. Thanks to the suggestion of the reviewer, we now also added a staining for CD31 and CD34. The new staining is included in the Figure S4 and is described in the lines 306-308: “We used antibodies against the vascular endothelial markers CD31 and CD34 to detect blood vessels and observed less, but bigger vessels in the tumor areas as compared to the surrounding brain parenchyma (Figure S4).”
The labels are now changed and have all a white background; we hope this modification made them more visible and clear.
- How long can you keep the cells in culture with no morphological changes and differentiation?
The cells have been kept in culture under spheroid-growing conditions for up to 60 consecutive passages with no obvious morphological changes. Differentiation has been observed only when cells were exposed to bovine serum (data not shown).
- A section for statistical analyses is missing in the M&M. Figure 1 needs statistics for example.
- We now added in the Material and Methods section a subsection called “4.6. Survival analysis” and we inserted, in Figure 1G and 1F, the statistic values of the Kaplan-Meier curves.
- Description of the survival studies (and Kaplan-Meyer) presented in Figure 1 is missing in the M&M.
- We now added in the Material and Methods section a subsection called “4.6. Survival analysis”
Reviewer 2 Report
This is a very novel and interesting manuscript where the researchers generated a panel of GBM cell lines through consecutive in vivo passaging of cells obtained from a Pten/p53 double-knockout mouse brain tumor model. They showed that a similar pattern of molecular heterogeneity as found in human GBM through transcriptome and genome analyses. After orthotopic transplantation these cells were able to mimic high-grade glioma with invasive and infiltrating features. These represent an exclusive model to further the field of GBM research.
Author Response
This reviewer did not request any change in the manuscript.
Reviewer 3 Report
Overall this is an excellent report of the generation and characterisation of a set of transplantable cell lines that generate GBMs in mice that closely mimick the genetic and phenotypic characteristics of human GBMs.
Authors have undergone a very solid work, from the initial design - successive passages of re-isolated cells/spheroids, re-implanted in C57/Bl6N mice. The authors have followed the mice for a long time (1 year) and characterised at the transcriptomic, genomic and histological level. They have also followed the animals with conventional MRI. There is also an immunohistochemical characterisation of the immune cell population. The work is sound, materials and methods, as well as data analyisis is sound and I could not identify any flaw in it.
I have only minor comments, in no particular order:
-Please correct some typos (Tumours cells, and other similar typos).
-Figure 1, make H&E pics bigger, there's space and the images would be better seen.
-I'd move supplementary figure 4 to the main document, I think that this result is really interesting. Where do the CD8s go? (they mention that in the Nat commun in press paper in the other model they hint that knowking out Sox1 then the tumours are more aggressive and have more infiltration from myeloid cells, look forward to reading this one to see which type of myeloid cells are these ones infiltrating, but just for the sake of comparison, I'd be interested in seeing which is the balance among the different immune cell populations, at least in the models presented in THIS paper, so the lymphocytes are also of interest to me and deserve to have their figure in the main document).
-Please include the number of animals studied.
-Authors mention that these lines are available to the scientific community, could they detail how to access them? I think that these lines could be of help for other groups working on the GL261 model and similar.
Author Response
Overall this is an excellent report of the generation and characterisation of a set of transplantable cell lines that generate GBMs in mice that closely mimick the genetic and phenotypic characteristics of human GBMs.
Authors have undergone a very solid work, from the initial design - successive passages of re-isolated cells/spheroids, re-implanted in C57/Bl6N mice. The authors have followed the mice for a long time (1 year) and characterised at the transcriptomic, genomic and histological level. They have also followed the animals with conventional MRI. There is also an immunohistochemical characterisation of the immune cell population. The work is sound, materials and methods, as well as data analysis is sound and I could not identify any flaw in it.
I have only minor comments, in no particular order:
-Please correct some typos (Tumours cells, and other similar typos).
- We checked the manuscript carefully and we corrected the typos that we identified.
-Figure 1, make H&E pics bigger, there's space and the images would be better seen.
- Following the suggestions of the reviewer we now increased the H&E sections in Figure 1A,B,C,D,E by 20%
-I'd move supplementary figure 4 to the main document, I think that this result is really interesting. Where do the CD8s go? (they mention that in the Nat commun in press paper in the other model they hint that knocking out Sox1 then the tumours are more aggressive and have more infiltration from myeloid cells, look forward to reading this one to see which type of myeloid cells are these ones infiltrating, but just for the sake of comparison, I'd be interested in seeing which is the balance among the different immune cell populations, at least in the models presented in THIS paper, so the lymphocytes are also of interest to me and deserve to have their figure in the main document).
- Following the suggestions of the reviewer, we now moved the Figure S4 into the main document as Figure 5. We are grateful to the reviewer for the indication, indeed we believe that glioma lymphocyte infiltration is an important information for a broader audience given the fact that our model can be used in a fully immunocompetent background.
-Please include the number of animals studied.
- We now added in the legend of Figure 1F and G and Figure S1A the number of animals used.
-Authors mention that these lines are available to the scientific community, could they detail how to access them? I think that these lines could be of help for other groups working on the GL261 model and similar.
- At the moment the cell lines are available by directly contacting Prof. Peter Angel. However, the authors are considering, upon publication of the manuscript, to deposit the cell lines in a publicly accessible repository.